# [Re] Deep Fair Clustering for Visual Learning

## Reproducibility Summary

**Scope of Reproducibility**

The authors propose a novel method for Deep Fair Clustering (DFC), combining existing frameworks for fair clustering—which typically have difficulty with high-dimensional large-scale data—with previous work on deep clustering—which typically has difficulty with fairness. Our reproducibility work targets the central claim that DFC learns fair representations with minimal utility loss and obtains superior results on both fairness and accuracy.

**Methodology**

We used the code repository made available by the authors and extended it to include support for pretraining, different datasets and comparative methods. We compare the DFC method against Deep Embedded Clustering (DEC) (11), which implements a comparable deep clustering method without fairness constraints, on the same four datasets (obtained from MNIST (7), USPS (6), MTFL (13) and Office-31 (9)) and fairness metrics that were used in the paper. We select one dataset (MNIST-UPS) for additional experiments aimed at validating the contribution of individual components of the DFC towards fairness. All experiments were run on a GeForce 1080Ti GPU. Hyperparameter optimization was performed using the Weights & Biases Sweeps feature (1).

**Results**

On the selected dataset, we reproduced accuracy to within 2% of reported value, normalized mutual information (NMI) and entropy to within 1%, and balance to within 5%. Our DFC method outperformed our DEC method on all accuracy and fairness metrics. We reproduced the accuracy of the non-digit datasets to within $1\%$ (Office-31) and $7\%$ (MTFL) but failed to obtain similar results for balance.

**What was easy**

We found no major challenges reproducing the provided code in as far as we used the author's provided pretrained models and the selected dataset (MNIST-USPS) used in the code.

**What was difficult**

Extending the code for the non-digit datasets was a challenge, as some hyperparameter settings and architecture details were difficult to infer from the paper. We found that performance was sensitive to small changes in the implementation and training of the encoders. Consequently, we ran into time and resource constraints when trying to reproduce all results for these datasets, due to the large number of models that required pretraining.

**Communication with original authors**

We had helpful one-off contact with the authors to verify hyperparameter settings.

# 1    Introduction

Machine learning (ML) is increasingly used in high-stake decision-making where the data contains sensitive attributes, such as gender, race or socioeconomic background. Examples include college admission, loan approval and bail/parole judgements. Such ML algorithms are vulnerable to bias and unfairness (4), and extensive literature has brought attention to the various challenges and inherent trade-offs in this field (5)(14).

Clustering is an important aspect of many such ML applications. To illustrate what it might mean for a clustering assignment to be unfair, we can consider its use in feature engineering, e.g., to label data in an unsupervised setting to increase its expressive power (2). In some cases, if the inherent structure of the training data differs between subgroups, standard clustering likely leads to partition assignments that correlate significantly with sensitive attributes. Often, these are controversial correlations that do not reflect true causal mechanisms. This could facilitate subsequent discrimination indirectly based on sensitive attributes, either knowingly or unknowingly.

In short, fair clustering is an ongoing, important, and complex field that still faces many challenges. The paper 'Deep Fair Clustering for Visual Learning' addresses various such challenges. Moreover, their method for fair clustering outperforms competitive fair clustering methods as well as competitive deep clustering methods. This shows that, if theoretically sound, their method is able to impose fairness constraints without significantly compromising clustering accuracy.

# 2    Scope of reproducibility

The paper aims to tackle the problem of fair clustering of high-dimensional data by introducing a Deep Fair Clustering (DFC) method suitable for image data. Their notion of fairness extends the demographic equipartition criterion and requires that the clustering assignment is independent of the protected subgroup membership. To achieve this independence, they leverage the feature representation encoding from deep methods and train a model to filter out sensitive attributes from the representations. Simultaneously, DFC enforces fairness constraints and optimizes clustering performance, notably with a minimal trade-off between the two. The central claims are summed up as follows:

- The first claim states that DFC achieves fair clustering of large-scale and high-dimensional visual data by learning fair representations of the input. To support this claim, we train DFC and DEC on the MNIST-USPS dataset and compare the learned representations in figure 3. The first row of table 1 shows the corresponding accuracy and fairness metrics of the final performance of both methods.

- The second claim regards the validity and effectiveness of the proposed minimax optimization formulation. In particular, we support this claim by recognizing DEC as DFC without the minimax optimization formulation (in particular, without the fairness components 4, 5 and the separate clustering of subgroups, see figure 1). To our best knowledge, this corresponds to the implementations in (11) and (8). The claim is supported by comparing DFC performance with DEC performance on accuracy and fairness metrics in table 1. In addition, figure 4 shows the performance of the discriminator during DFC, which provides some insight in the extend to which DFC achieves 'masked' representations during trainig.

- The third claim states that DFC shows superior performance on four real-world visual datasets (see 3.1). We aim to show this by running experiments on the same four datasets with both DEC and DFC and compare performance in table 1.

# 3    Methodology

Each claim requires us to implement the DFC method and the DEC method and train it on the MNIST-USPS dataset. From here, we address each claim separately in three sets of experiments. For the first, we visualize the representations using dimensionality reduction methods and compare the obtained clusters. For the second, we compare the DEC and DFC on accuracy, NMI, Balance and Entropy metrics. Moreover, we monitor discriminator accuracy for DFC. For the third, we run our experiments with the three other datasets that are used in the paper and compare results on the above mentioned four metrics.

Before discussing the details, we briefly explain the DFC method as we implemented it based on the authors' description. The method has several components that are trained simultaneously in a minimax optimization scheme. An overview of the method is shown in figure 1. The task is to cluster datapoints $X$ with sensitive categorical attribute $G$ belonging to protected subgroup $G(X) \in [M]$ into $K$ clusters. A feature encoder $\mathcal{F}(X)$ transforms the data $X$ into representations $Z$. As the figure shows, each subgroup is clustered separately and requires a separate, pretrained encoder. The clustering assignment $P = \mathcal{A}(Z)$, taken from the Deep Embedded Clustering method (DEC), creates a soft assignment $P$

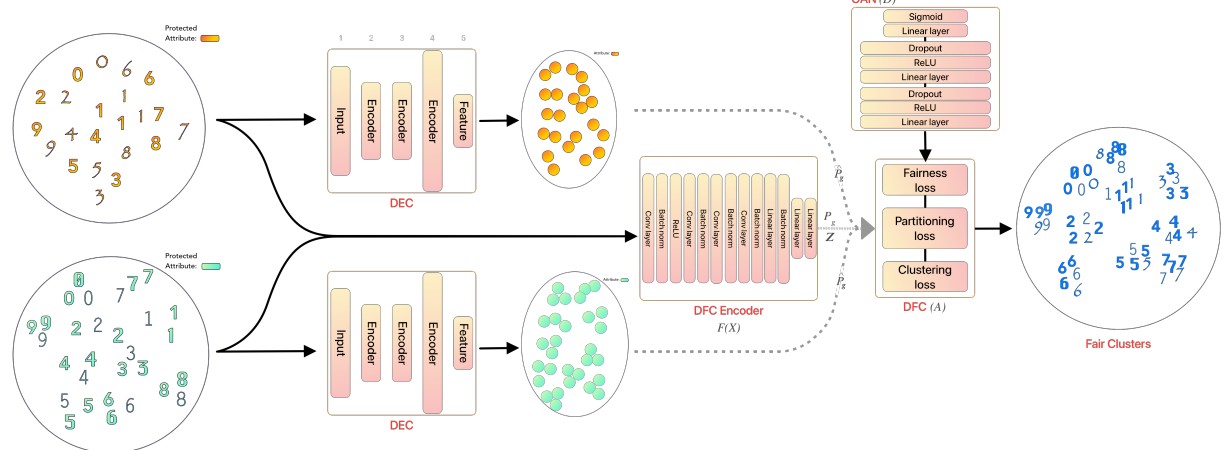

Figure 1: Overview of Deep Fair Clustering. The orange and green colors represent the protected subgroups.

reflecting the probability of assigning datapoints to each cluster, once for each protected subgroup, and once for the full dataset.

A discriminator $\mathcal{D}$ aims to reconstruct the protected subgroup membership based on the soft assignment $P$

**Encoder**    Central to the DFC approach is the encoding of fair representations. Transforming the input data using feature encoding is important, in the first place, because (deep) clustering at pixel level does not often yield good results. In DFC, the encoded representations are at the same time an important step towards fairness. The authors note that "fair representations are achieved when the discriminator cannot distinguish between representations from different protected subgroups." To this end, the encoder is finetuned simultaneously with the rest of the clustering algorithm.

**Discriminator**    A discriminator is constructed to monitor and guide the fairness of representations $Z$.

**Deep Embedded Clustering**    For the clustering assignment, the authors closely follow the DEC method. As in (11), the cluster assignment is learned using Stochastic Gradient Descent (SGD), repeating the following two processes:

1. A soft assignment of the datapoints to cluster centroids, using the Student's t-distribution as a kernel to measure the similarity between embedded point $z$ and centroid $c_k$:

$$p_k = \frac{(1 + ||z - c_k||^2/\alpha)^{-\frac{\alpha+1}{2}}}{\sum_{k'}(1 + ||z - c_{k'}||^2/\alpha)^{-\frac{\alpha+1}{2}}}, \tag{1}$$

   where $p_k$ is the probability of a datapoint belonging to cluster center $c_k$ and $\alpha$ the degrees of freedom of the Student's t-distribution.

2. The obtained distribution $P$ of soft assignments is matched to an auxiliary distribution Q:

$$q_k = \frac{(p_k)^2/\sum_{x \in X_g} p_k}{\sum_{k' \in [K]}((p_{k'})^2/\sum_{x \in X_g} p_{k'})}, \tag{2}$$

   which is calculated for each protected subgroup separately. Finally, the clustering is optimized by minimizing the KL-divergence loss:

$$\mathcal{L}_c := KL(P||Q) = \sum_{g \in [M]} \sum_{x \in X_g} \sum_{k \in [K]} p_k \log \frac{p_k}{q_k}. \tag{3}$$

   This ensures that the soft assignment matches the target distribution as closely as possible.

**Deep Fair Clustering**    DFC implements DEC as above, but adds two loss functions to impose the fairness constraint while promoting cluster utility. To encourage fair partitions according to the demographic parity, the authors introduce a *fairness adversarial loss*:

$$\mathcal{L}_f := \mathcal{L}(\mathcal{D} \circ \mathcal{A} \circ \mathcal{F}(X), G), \tag{4}$$

Where $\mathcal{L}$ is the cross-entropy loss function and $\circ$ denotes function composition, in particular the result of the encoder $\mathcal{F}(X)$, the clustering assignment $\mathcal{A}$ and the discriminator $\mathcal{D}$ applied to the data in sequence. The discriminator $\mathcal{D}$ is

implemented as a Generative Adversarial Network (GAN) (see figure 1) with the exact inverse architecture of the used encoder. It monitors how well the DFC is able to reconstruct the protected subgroup based on the predicted cluster probabilities of a datapoint. With respect to the *fairness adversarial loss*, $\mathcal{D}$ is maximized with the probability of assigning the correct membership label for each sample. Simultaneously, clustering $\mathcal{A}$ and representations $\mathcal{F}(X)$ are trained to maximally confuse $\mathcal{D}$.

To increase the clustering performance under the fairness constraint, DFC adds a *structure preservation loss*:

$$\mathcal{L}_s = \sum_{g \in [M]} \left\| \hat{P}_g \hat{P}_g^\top - P_g P_g^\top \right\|^2, \tag{5}$$

where $\hat{P}_g$ is the soft assignment for a datapoint in protected subgroup $g$, and $P_g$ is the result for subgroup $g$ when the clustering was performed over the complete dataset. In other words, based on theoretical considerations as described in the original paper (8), DFC expects local and global subgroup clustering partition to be similar.

The final learning objective is as follows:

$$\max_{\mathcal{F}, \mathcal{A}} \quad \alpha_f \mathcal{L}_f - \alpha_s \mathcal{L}_s - \mathcal{L}_c \tag{6}$$

$$\min_{D} \quad \alpha_f \mathcal{L}_f \tag{7}$$

Note how the fairness adversarial loss (4) is simultaneously maximized w.r.t. $\mathcal{F}$ and $\mathcal{A}$ (6) to minimize performance of the discriminator as a result of clustering, and minimized w.r.t $\mathcal{D}$ (7) to optimize its parameters and obtain a saddle-point solution. The values for $\alpha$ are 1 by default. Following the paper, for the office-31 dataset $\alpha_s$ is recalculated as $(512/128)^2 \cdot (31/10)$ to adjust for difference in batch size and cluster number.

### 3.1 Datasets

We perform our experiments on the same four datasets as in the original paper. To test claim 1 and 2, we restrict ourselves to the *MNIST-USPS* dataset, obtained from combining the *MNIST* dataset (7) and the *USPS* dataset (6). From both, only the train images are taken, 60,000 and 7,261 respectively. The *Color Reverse MNIST* is obtained by reversing the black and white pixels in the MNIST dataset and combining the normal and the reversed version. The *The Multi-task Facial Landmark (MTFL)* dataset (13) consist of facial recognition pictures, from which 1000 images with and 1000 without glasses are sampled randomly. Finally, the *The Office-31* dataset (9) consists of images from 31 different categories collected from two distinct domains: Amazon and Webcam. For important statistics, see figure 2.

| Dataset | Type | # Instances | # Classes | # Dim. | Sensitive Attribute | Max of Bal. | Max of Ent. |
|---|---|---|---|---|---|---|---|
| MNIST-USPS | digital | 67,291 | 10 | 1,024 | source of digital | 0.120 | 2.303 |
| Color Reserve MNIST | digital | 120,000 | 10 | 1,024 | original or reversed | 1.000 | 2.303 |
| MTFL | face | 2,000 | 2 | 50,176 | with or w/o glasses | 1.000 | 0.693 |
| Office-31(*A&W*) | object | 3,612 | 31 | 50,176 | domain source | 0.273 | 3.434 |

Figure 2: Characteristics of each of the four datasets, taken from the original paper (8).

#### 3.1.1 Hyperparameters

The DFC model is dependent on 5 pretrained models; a VAE and a DEC for each subgroup and a VAE for the whole dataset. Since the optimal hyperparameters of the DFC are likely to depend on the hyperparameter choices of the pretrained models, a complete search would be expensive. To make hyperparameter search manageable we make the assumption that the optimal hyperparameters are independent of the input data and the pretrained models it uses. This assumption enables to search the hyperparameter space of the DFC, DEC and VAE independently. Note that the VAE only needs to be pretrained and optimized for the MNIST-USPS and Reverse MNIST datasets, the other datasets use a pretrained ResNET50 encoder.

The hyperparameter search for the DEC and the VAE focused on finding the best combination of the learning rate, batch size and the number of epochs. These searches were performed with Bayesian optimization and contained respectively 32 and 18 experiments on MNIST. The VAE Bayesian search was set to maximize the accuracy of the KMeans clustering afterwards, which is also used in the training of the DFC as initial centroids. The goal of the DEC was to optimize its clustering accuracy.

The DEC and VAE were pretrained using the optimal parameters on MNIST and USPS to facilitate the hyperparameter search of the DFC. The focus was on finding the optimal learning rate, batch size, and number of epochs. A random grid search was used because it is not clear which metric should be optimized. The best performing set of parameters was chosen based on the Accuracy, NMI, entropy and balance.

The range of the hyperparameter space was inspired by hyperparameters mentioned by the authors in the paper, email contact and a DEC MNIST example[1].

The hyperparameters of the KMeans clustering were set to the same values the authors used. TSNE hyperparameters were set to defaults as can be seen in Appendix C, which contains a complete list of all hyperparameters and overviews of the hyperparameter searches.

## 4  Experimental setup and implementation

All experiments were run on cuda enabled computers with seeds set to 2019 before training each model, unless otherwise specified. Our code containing the experiments is available at our GitHub repo [2]. Our code is strongly based on the implementation provided by the authors [3]. We have found that their code is a clear reflection of the paper. However, it was necessary to extend their implementation to provide support for pretraining, other datasets and other methods.

The DFC architectures described in the paper relies on several models joint together. Roughly, it includes an encoder, a clustering assignment and a discriminator. Some experiments require slightly different model combinations, which is always made explicit.

### 4.1  Pretrained Encoder

The type of encoder can be chosen independently from the DFC. The authors use different architectures based on the complexity of the dataset; for the digit based datasets a standard Variational Autoencoder is used, for the other datasets a pretrained ResNet50.

The standard VAE is based on four blocks of a 2D convolutional layer, batch normalization and a ReLU activation layer. This is scaled down in 2 steps with fully connected layers to the 64 dimensional hidden space. Since the decoder was missing from the author's implementation, some guesses had to be made about the exact implementation, we tried to follow conventions as much as possible. A standard KL-Divergence and MSE loss were used for pretraining. The exact architecture can be found in Appendix A.

For MTFL and Office-31, considering their complex and high resolution images, the authors use ImageNet-pretrained ResNet50 (3). The authors did not mention how the ResNet is finetuned exactly. Due to the relatively small size of the datasets used for the experiments, it was decided to only finetune the last layer during training the DFC.

### 4.2  Pretrained DEC

The structural preservation loss, discussed before 5, is calculated using two separate DEC models. These models could be seen as the golden standard for each subgroup. Since they are equally weighted, this ensures that the protected subgroup is underrepresented in the loss. These DECs also require an encoder to transform the images. The same encoder training methods and architectures are used for all encoders. The DECs were implemented as a stripped down DFC with three modifications. The fairness and the structural preservation loss were removed. Lastly, the clustering loss is calculated over the entire batch at once, instead of individually per subgroup. This results in an exact implementation of the original DEC paper (11).

### 4.3  Centroid Initialization

The deep clustering methods DEC and DFC both benefit from properly initializing the Student's t-distribution cluster definitions. This is done with a simple KMeans clustering approach. KMeans is trained on the entire dataset that is used by the clustering method afterwards. The final centroids learned are copied to be the initial cluster definitions.

---

[1] https://github.com/vlukiyanov/pt-dec/blob/master/examples/mnist/mnist.py

[2] https://github.com/Joppewouts/Reproducing-Deep-Fair-Clustering

[3] https://github.com/brandeis-machine-learning/DeepFairClustering

### 4.4 DFC

The DFC consists of a pretrained encoder, a discriminator module and cluster assignment module. The weights of these modules are updated with the Adam optimizer, using an inverse exponential learning rate scheduler. The discriminator has a standard architecture, details are attached in Appendix B. As described by the authors, the learning rate of the discriminator uses a multiplication factor of 10.

#### 4.4.1 Clustering Assignment

The fairness constraints and learning objectives of DFC rely on clustering being performed on the subgroups separately. Since these subgroups are homogeneous with respect to sensitive attributes, they can be clustered irrespective of fairness-considerations. Therefore, the same model can be used as in the Deep Embedding clustering. Accordingly, the authors use the DEC clustering for the subgroups separately, as described in section 3. For full details see also (11).

### 4.5 Competitive methods

Due to time constraints, we chose to only implement one competitive method. DEC was a compelling candidate because it allows to compare fair deep clustering with ordinary deep clustering.

### 4.6 Metrics

Four metrics are used to evaluate performance; accuracy, normalized mutual information, balance and entropy. These are the same metrics as in the original paper, which includes detailed explanations. Furthermore, for evaluating the fairness two more metrics are implemented. The accuracy of the discriminator is computed to function as a proxy for the fairness of the cluster assignments. The probabilities outputted by the discriminator are converted to subgroup classes by thresholding at $0.5$. An accuracy of $50\%$ indicates fully masked representations, and therefore fair cluster assignments. Secondly, TSNE (10) is implemented to visualize the latent space learned by the encoder. In an ideal fair representation, the representation of the two subgroups should be indistinguishable. For example, the digit 5 of MNIST and the digit 5 of USPS both encoded as the character '5' would be ideal, since the representations are indistinguishable. The goal of a TSNE mapping is that distant samples in the high dimensional space are also far away from each other in the lower dimension, and vice versa. Therefore, in a visualization of a fair latent space, it should be hard to distinguish clusters of protected subgroups. All results are calculated over 5 different seeds: 0, 1, 42, 2019 and 2020. However, it was not possible to provide the variance for the results that require a ResNet encoder due to time and resource limitations.

#### 4.6.1 Computational requirements

We trained all models on an 8.25MB GeForce 1080Ti GPU. Pretraining VAE encoders for each dataset took $(5\times) \pm$ 45min, pretraining the DEC models $(7\times) \pm$ 1hr. The sweep for optimal parameter search took $(20\times)$ 1hr and running the DFC and DEC models for all four datasets on 5 random seeds took $(8\times) \pm$ 1hr GPU time. The total experiments required a total of $\pm$40hr GPU time.

## 5 Results

Our results support the first claim that DFC learns fair representations that do not cause much loss in utility and the second claim that the minimax optimization formulation contributes to clustering that is both fair and accurate. We reproduced accuracy on the MNIST-USPS dataset to within 2% of reported value, normalized mutual information (NMI) and Entropy to within 1%, and balance to within 5%. Our DFC method outperformed our DEC method on all accuracy and fairness metrics. We reproduced the accuracy of the Office-31 datasets to within $1\%$ and the MTFL dataset to within $7\%$, but failed to obtain similarly good results for the fairness metrics.

### 5.1 Claim 1: Learning fair representations for the MNIST-USPS dataset

Figure 3 shows the learned representation of the DFC method (top) and the DEC method (bottom). Protected subgroup membership is indicated by color for MNIST (red) and USPS (blue). Monitoring the DFC discriminator performance shows that the accuracy for recovering the protected subgroup drops from $60\%$ at the first 1000 runs to $50\%$ after 2000 iterations. The corresponding accuracy and fairness metrics are shown in table 1. We obtain an accuracy within $2\%$ of the accuracy reported in the paper. NMI and Entropy scores fell within $1\%$ of the original findings, and balance was $5.5\%$ lower in our experiments.

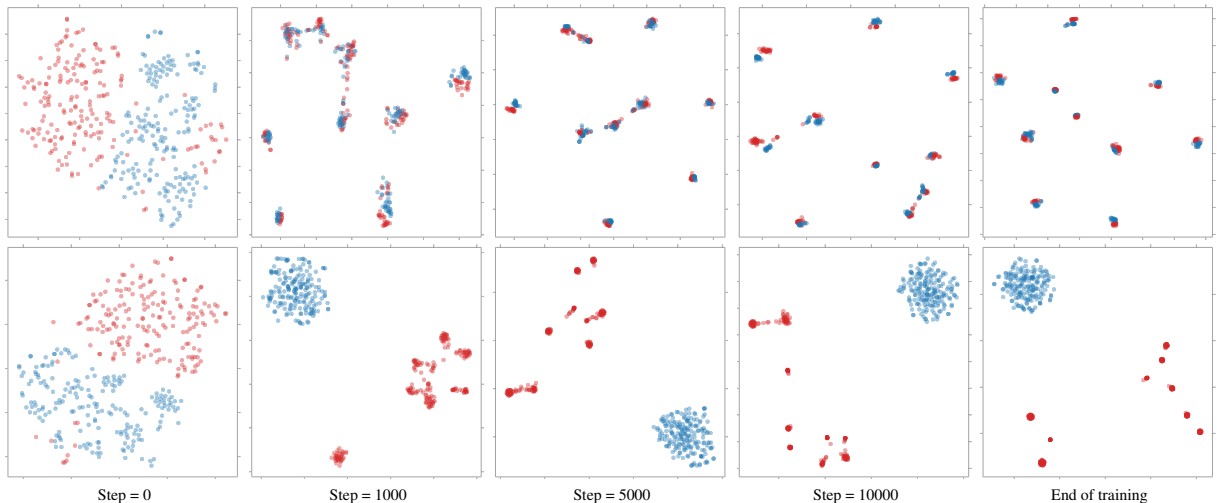

| Step = 0 | Step = 1000 | Step = 5000 | Step = 10000 | End of training |

Figure 3: TSNE visualizations of the latent space learned by the encoder on MNIST-USPS with the final models. The rows respectively show the representations for the DFC and for the DEC. The DFC learns representations of the images which results nicely in 10 clusters in the visualized space. The DEC learns 9 distinct representation for MNIST digits and does not seem to separate USPS digits at all.

## 5.2 Claim 2: Validity of minimax optimization formalization

The first two rows of table 1 show performance on all four metrics for the DFC and DEC methods. Results indicate that DFC outperforms DEC both in accuracy and in fairness methods. Moreover, the table shows that our implementation of the DEC method achieves a $17\%$ higher accuracy and performs better than the original implementation of the DEC on all other metrics except balance.

To further investigate how subcomponents of DFC contribute towards performance and fairness, we plot the decoder accuracy in figure 4. Interestingly, the plot shows that DFC initially achieves perfectly masked representation up to around $8.5k$ training steps, where $\mathcal{F}(X)$ and $\mathcal{A}$ no longer succeed in fully confusing $\mathcal{D}$, presumably in favour of cluster favourable representations. This indicates that some trade-off is still present.

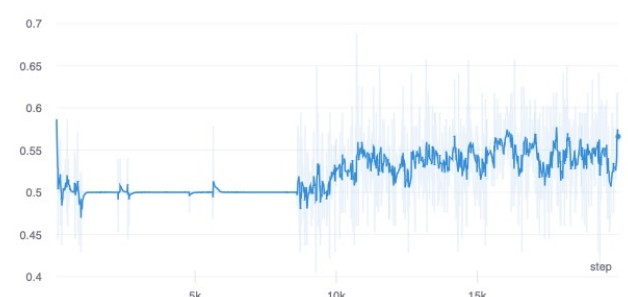

Figure 4: DFC Discriminator accuracy over training steps. An accuracy of 0.5 indicates that representations cannot be discriminated for subgroup membership.

## 5.3 Claim 3: Different datasets

The results for all datasets are can be found in table 1. Our implementation of DFC outperformed DEC on the three other datasets and showed similar accuracy as reported in the original paper. Results for entropy were also consistently higher than the DEC implementations and similar to the entropy reported in the original paper. The balance, however, was much lower than in the original paper, and did not differ significantly from the DEC results.

## 6 Discussion

For the MNIST-USPS digit datasets, the results indicate that the method is able to learn fair representations and clusters. The entropy results indicate equal fair partitions, the TSNE visualizations show good clusters and the discriminator is not able to distinguish between subgroups in the cluster assignments.

We were partly unable to reproduce its fairness performance on the three other datasets but can think of various limitations of our approach that might provide an explanation for this. First, we put notable effort into adjusting and training pretrained models for the MNIST-USPS experiments, and performed a hyperparameter search to find

| Dataset | | Method | Acc | | NMI | | Balance | | Entropy | |
|---|---|---|---|---|---|---|---|---|---|---|
| MNIST-USPS | ours | DFC | 0.81 | (0.012) | 0.80 | (0.015) | 0.012 | (0.001) | 2.292 / 2.299 | (0.004 / 0.001) |
| | | DEC | 0.76 | (0.004) | 0.73 | (0.003) | 0.000 | (0.000) | 2.166 / 2.278 | (0.007 / 0.002) |
| | orig. | DFC | 0.83 | | 0.79 | | 0.067 | | 2.301 / 2.265 | |
| | | DEC | 0.59 | | 0.69 | | 0.000 | | 2.082 / 1.735 | |
| Reverse MNIST | ours | DFC | 0.39 | (0.027) | 0.37 | (0.038) | 0.007 | (0.001) | 2.283 / 2.293 | (0.009 / 0.002) |
| | | DEC | 0.32 | (0.010) | 0.32 | (0.018) | 0.000 | (0.000) | 1.395 / 1.939 | (0.118 / 0.055) |
| | orig. | DFC | 0.58 | | 0.68 | | 0.763 | | 2.294 / 2.301 | |
| | | DEC | 0.40 | | 0.48 | | 0.000 | | 1.774 / 1.384 | |
| MTFL | ours | DFC | 0.79 | | 0.27 | | 0.001 | | 0.655 / 0.693 | |
| | | DEC | 0.77 | | 0.23 | | 0.001 | | 0.305 / 0.693 | |
| | orig. | DFC | 0.72 | | 0.19 | | 0.986 | | 0.693 / 0.693 | |
| | | DEC | 0.52 | | 0.03 | | 0.711 | | 0.660 / 0.576 | |
| Office 31 | ours | DFC | 0.68 | | 0.72 | | 0.000 | | 2.689 / 3.393 | |
| | | DEC | 0.59 | | 0.65 | | 0.000 | | 2.300 / 3.079 | |
| | orig. | DFC | 0.69 | | 0.72 | | 0.117 | | 3.422 / 3.403 | |
| | | DEC | 0.55 | | 0.60 | | 0.000 | | 3.063 / 2.937 | |

Table 1: Cluster performance and fairness results compared to the original paper with the variance over 5 seeds when applicable.

optimal settings. Due to time and resource constraints, we were not able to repeat this process for all datasets, which may have led to different results. Secondly, our use of the ResNet50 encoder differed slightly from that in the paper. Correspondence with the authors had confirmed that their implementation finetuned all parameters of ResNet50. However, for us GPU memory constraints made it impossible to store all gradients that would be required. Therefore, we resolved to only finetune the final layer, based on transfer learning literature (12) and the relatively small sizes of the datasets. This could partially explain the difference in results.

However, evaluating our full results, an additional question about the added fairness constraints and the experiment setup of this method arose. In the case of MNIST-USPS, optimal clustering solutions for the datasets closely match optimal fairness according to the constraints. Since the digit datasets are quite nicely balanced in the sense that they have similar partitions across subgroups, the equal partition definition of fairness is in line with the optimal solution, and we suspect that the fairness constraints here do not pose as much of a challenge as they might in more imbalanced datasets. For this reason, we were particularly interested in the results for the non-digit datasets. Further research might be needed to investigate the theoretical framework in the context of datasets with different inherent fairness challenges.

## 6.1 What was easy and/or difficult

Based on the implementation made availabe by the authors it was straightforward to evaluate the method on MNIST-USPS, using the provided pretrained encoders, DECs and initial centroids. These results matched the results achieved in the original paper. However, essential code and specifications were missing to extend these results to other datasets and methods. Examples include the hyperparameters for pretraining all the models, hyperparameters for running on other datasets, missing specifications for the Decoder of the VAE and the finetuning of ResNet50. This made it hard to replicate the method for other datasets.

Despite the missing parts, the code was helpful for understanding the method in detail. Most importantly, it cleared up the workings of the structural preservation loss which was not immediately clear from the paper.

The large number of different pretrained models needed was another difficulty. The DFC for MNIST-USPS depends on 8 other models: 3 encoders, 2 DECs and 3 KMeans cluster initializations. This, in combination with the fact that the DFC is sensitive to the quality of the pretrained models, made it hard to debug performance issues that could be caused by the model itself or any of the previous methods. In addition, it required excellent experiment management to keep track of which models are trained with which pretrained models and parameters.

## 6.2 Communication with original authors

We consulted the authors about hyperparameters for pretraining the encoders and which layers of ResNet50 to finetune. We received a helpful response in which our questions were adequately answered.

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

# Appendices

## A  Variational Autoencoder Architecture

| Layer Name | Output Size | Settings |
|---|---|---|
| conv1 | $16 \times 32 \times 32$ | filter: $3 \times 3$; stride: 1; padding: 1 |
| bn1 | $16 \times 32 \times 32$ | features: 16 |
| ReLU | $16 \times 32 \times 32$ | |
| conv2 | $32 \times 16 \times 16$ | filter: $3 \times 3$; stride: 2; padding: 1 |
| bn2 | $32 \times 16 \times 16$ | features: 32 |
| ReLU | $32 \times 16 \times 16$ | |
| conv3 | $32 \times 16 \times 16$ | filter: $3 \times 3$; stride: 1; padding: 1 |
| bn3 | $32 \times 16 \times 16$ | features: 32 |
| ReLU | $32 \times 16 \times 16$ | |
| conv4 | $16 \times 8 \times 8$ | filter: $3 \times 3$; stride: 2; padding: 1 |
| bn4 | $16 \times 8 \times 8$ | features: 16 |
| ReLU | $16 \times 8 \times 8$ | |
| Flatten | 1024 | |
| fc1 | 512 | |
| bn5 1D | 512 | features: 512 |
| ReLU | 512 | |
| **mu** | 64 | |
| **logvar** | 64 | |

Table 2: Encoder architecture. The convolutional and batchnorm layers are standard 2D versions unless otherwise indicated. The last two layers are the output layers for respectively the mean and log variance of the distribution. These both take the last ReLU layer as input. Batch size dimension is omitted.

| Layer Name | Output Size | Layer |
|---|---|---|
| fc1 | 512 | |
| ReLU | 512 | |
| fc2 | 1024 | |
| bn1 1D | 1024 | features: 1024 |
| ReLU | 1024 | |
| UnFlatten | $16 \times 8 \times 8$ | |
| convT1 | $32 \times 16 \times 16$ | filter: $3 \times 3$; stride: 2; padding: 1; output padding: 1 |
| bn2 | $32 \times 16 \times 16$ | features: 32 |
| ReLU | $32 \times 16 \times 16$ | |
| convT2 | $32 \times 16 \times 16$ | filter: $3 \times 3$; stride: 1; padding: 1; output padding: 0 |
| bn3 | $32 \times 16 \times 16$ | features: 32 |
| ReLU | $32 \times 16 \times 16$ | |
| convT3 | $16 \times 32 \times 32$ | filter: $3 \times 3$; stride: 2; padding: 1; output padding: 1 |
| bn4 | $16 \times 32 \times 32$ | features: 16 |
| ReLU | $16 \times 32 \times 32$ | |
| convT4 | $1 \times 32 \times 32$ | filter: $3 \times 3$; stride: 1; padding: 1; output padding: 0 |
| bn5 | $1 \times 32 \times 32$ | features: 1 |
| Sigmoid | $1 \times 32 \times 32$ | |

Table 3: Decoder architecture. Transposed Convolutions, indicated with *convT*, are used as a inverse of standard convolutional layers. The architecture was chosen to resemble the inverse of the encoder. The final non linear activation is a sigmoid since this matches the loss function and original image value range. Batch size dimension is omitted.

# B   Discriminator Architecture

| Layer Name | Output Size | Layer |
|---|---|---|
| fc1 | 32 | |
| ReLU | 32 | |
| Dropout | 32 | Keep prob: 0.5 |
| fc2 | 32 | |
| ReLU | 32 | |
| Dropout | 32 | Keep prob: 0.5 |
| fc3 | 1 | |
| Sigmoid | 1 | |

Table 4: Discriminator architecture. The input are cluster assignment per sample of shape $B \times C$, where $B$ is the batch size and $C$ the number of clusters.

# C   Hyperparameters

| Model | Parameter | Range | Optimal VAE | Optimal ResNet |
|---|---|---|---|---|
| **VAE** | batch size | $64, 128, 256$ | 64 | - |
| | learning rate | 1e-4, 1e-3, 1e-2, 1e-1 | 0.001 | - |
| | epochs | $20, 50, 100, 500$ | 50 | - |
| **DEC** | batch size | $64, 128, 256$ | 256 | 128 |
| | iterations | $10000 - 60000$ | 50000 | 20000 |
| | learning rate | 1e-4, 1e-3, 1e-2, 1e-1, 1, 10 | 0.0001 | 0.0001 |
| **DFC** | batch size | $64, 128, 256$ | 64 | 128 |
| | iterations | $20000, 35000, 50000$ | 20000 | 20000 |
| | learning rate | 1e-4, 1e-3, 1e-2, 1e-1 | 0.001 | 0.001 |

Table 5: Complete list of hyperparameter search space and the optimal values. The search was performed with bayesian and a grid search for the datasets that rely on the custom VAE. Because of resource limitations the hyperparameter search for the more complex datasets that rely on ResNet, the search was manually and less extensive performed.

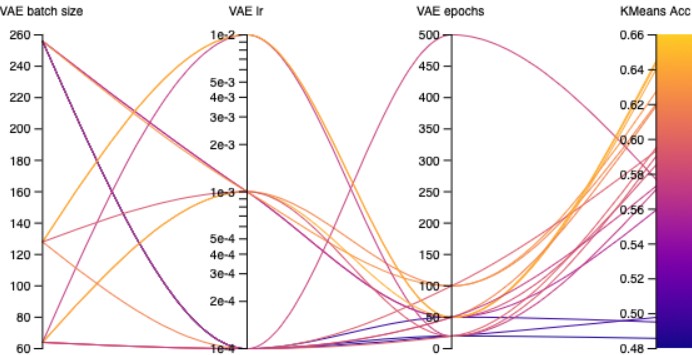

Figure 5: Overview of the results for the VAE hyperparameter search, evaluated with the clustering performance of a KMeans module on MNIST-USPS.

| Model | Parameter | Value |
|---|---|---|
| **KMeans** | n init | 20 |
| | max steps | 5000 |
| | init | k-means++ |
| | tol | 1e-4 |
| | precompute distances | auto |
| | algorithm | auto |
| **TSNE** | n components | 2 |
| | perplexity | 30 |
| | early exaggeration | 12 |
| | learning rate | 200 |
| | n iter | 1000 |
| | metric | euclidean |
| | init | random |
| | method | barnes hut |
| | angle | 0.5 |
| **VAE & DEC** | Adam betas | $(0.9, 0.999)$ |
| | Adam weight decay | 0 |
| **DFC** | Adam betas | $(0.9, 0.999)$ |
| | Adam weight decay | 1e-4 |

Table 6: Complete list of default hyperparameters per model not included in the hyperparameter search.

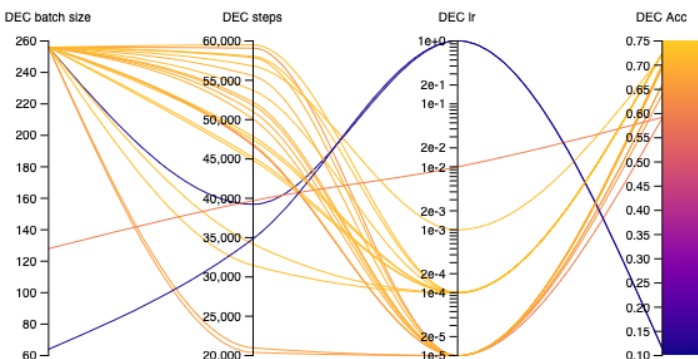

Figure 6: Overview of the results for the DEC hyperparameter search using the best performing pretrained VAE on MNIST-USPS.

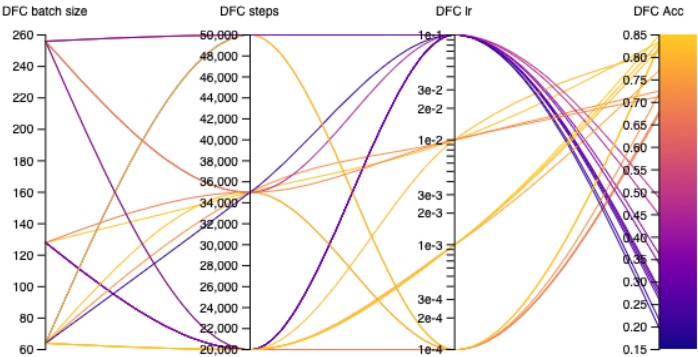

Figure 7: Overview of the results for the DFC hyperparameter search using the best performing pretrained VAE and DEC evaluated on MNIST-USPS.

