# OpenReview forum: "Reproducibility report: Deep Fair Clustering for Visual Learning"
_ML_Reproducibility_Challenge/2020 — Reject_

### Official Review · AnonReviewer1 · 2021-03-01
**Good reproduction of original work**

**Rating:** 7
**Confidence:** 4

**Review:**

**Quality:** Overall quality of the reproduction is very good. The writing reveals the good understanding of the original work by the authors of the reproducibility report.

**Clarity:** I found the report to be written very clearly. The ease and difficulty faced by the authors is clearly mentioned in the report. The exact reproductions, understandings and explanations are also conveyed very clearly, both verbally and pictorially.

**Originality:** The submitted report is quite original on its own. The authors put their own knowledge in the domain to judge and reproduce the original work. The thought process of the authors are distinctly visible.

**Significance:** The report provides good insights on how the experiments in the original paper actually work, while also generating new hypothesis to be tested for future research, which is a positive outcome.

**Pros**

- detailed reporting of every aspect
- clear listing of shortcomings in the original paper
- thoughtfulness of the authors to present the reasons of unmatched results

**Cons**

- detailed labelling within figures found missing, making it difficult to understand the diagrams in first glance

**Familiar With The Original Paper:**

I have read the original paper

**Reproducibility Summary:**

Report has summary

---

### Official Review · AnonReviewer3 · 2021-03-01
**Replication study seems to done right**

**Rating:** 6
**Confidence:** 5

**Review:**

The replication study pretty much stuck to the assumptions, code and interpretations of the original paper. Unsurprisingly, they found issues with context -sensitive aspects of the study -- which do not have solid theoretical foundations. They found issues with replicating the results on two datasets -- this part appears to be done correctly. All this suggests that related theoretical frameworks need more work.

**Familiar With The Original Paper:**

I have read the original paper

**Reproducibility Summary:**

Report has summary

---

### Official Review · AnonReviewer2 · 2021-03-08
**Good report but metrics need to be verified/recaliberated**

**Rating:** 7
**Confidence:** 4

**Review:**

### SUMMARY

This work offers an in-depth exploration of the reproducibility of "Deep Fair Clustering for Visual Learning" which claims that:
- The proposed method ensures cluster validity and fairness on large-scale, high-dimensional visual learning while (a) seeking to find feature mappings amenable for structure discovery and (b) filtering out sensitive attributes.
- The process has been modeled as a minimax optimization problem where cluster analysis and assignment is independent of sensitive attributes (aka C(X) is statistically independent of G) with minimal utility loss, high accuracy score and fairness measure.
- The theoretical analysis has been backed by empirical demonstration on four visual datasets: MNIST-USPS, MTFL, Color Reverse MNIST, Office-31 using the following metrics: Accuracy, Normalised Mutual Information, Balance and Entropy.

The submitted report addresses the above claims as follows:

- Visualizes learned representations of encoder through t-SNE to demonstrate DFC's ten clearly separated clusters ensuring fairness.
- Considers DEC as DFC without minimax optimization (in particular without - separate subgroup clustering, fairness adversarial loss and structural preservation loss).
- Comparisons across DEC and DFC have been verified across the specified metrics (not for the following models stated in the original paper: DAC, AE, CIGAN, ScFC, SpFC, FCC, and FAlg) for the above mentioned datasets. This report further claims to extend the original code repository to include support for pre-training, different datasets, comparative methods with additional hyperparameter optimization using the Weights & Biases Sweeps feature.

### MERITS
The report is well-written and intrinsic details of the original paper are expounded to reasonable depth. Results have been reproduced satisfactorily.

### RECOMMENDATIONS
- In Figure 1, it might be helpful to incorporate notations (For Example: feature encoder F(X) transforms data into Z) into the schematic representations as well.
- Statistics in Section 3.1 can be discussed in a tabular form. As per the [Machine Learning Reproducibility Checklist](https://www.cs.mcgill.ca/~jpineau/ReproducibilityChecklist.pdf), I'd also recommend including links to downloadable versions of the dataset for Office-31, Color-reverse MNIST and MTFL.
- Results of the original paper on the Balance metric of Office-31 dataset has not been reported accurately: Please recheck. The reproducibility report may need to be updated accordingly. Also, re-verify your reported percentage intervals for various metrics.
- Regarding the claim “Additional experiments aimed at validating the contribution of individual components of the DFC towards fairness”, please elucidate the experimentation as well as individual contributions of the following major components (DEC/Discriminator/Encoder). These seem to have been (a) cryptically addressed in the report and (b) contributions have been fragmented across sections (in the reviewer's opinion) and hence, do not bolster your argument.
- I'd also recommend including the table of results into the README document on [your github repository](https://github.com/Joppewouts/Reproducing-Deep-Fair-Clustering). Optionally, you could include the list of dependencies in the report.

### MINOR CORRECTIONS
- In Appendix A, Table 2 Caption: "meand and log variance of the distribution"; **Correction:** "_mean_ and log variance of the distribution"
- In Appendix A, Table 3 Caption: "The architecture was chosen to resemble te inverse of the encoder"; **Correction:** "The architecture was chosen to resemble _the_ inverse of the encoder".
- In Section 3, Deep Embedded Clustering, Line 89: "A soft assignement of the datapoints to cluster centroid"; **Correction:** "A soft _assignment_ of the datapoints to cluster centroid".
- In Section 3, Deep Fair *Clustering*, Line 100: "an fairness adversarial loss"; **Correction:** "_a_ fairness adversarial loss". (_Clustering_ instead of _clustering_ to ensure consistency of casing across headings/sub-headings)
- In Section 3, Deep Fair Clustering, Line 110:  "In other words, they expect, given a subgroup, that the clustering result is similar when trained only on the subgroup data and when trained on all subgroups and looking only at the samples of one subgroup, and encourage the distributions to be similar"; I believe this statement can be better phrased in terms of local subgroup vs global subgroup distribution and clustering.
- In Section 3.1, “The The Multi-task Facial Landamark (MFTL)”; Line 129: “_The_ Multi-Task Facial _Landmark_”, "constist of 12,995 face images". **Correction:** "_consist_ of 12,995 face images"
- In Section 4, Experimental Setup and Implementation, Line 165: "Some experiments require sightly different model combinations"; **Correction:** "Some experiments require _slightly_ different model combinations''
- In Section 4.3, Centroid Initialization, Line 188: "student's t-dsitribution"; **Correction:** "student’s _t-distribution_"
- In Section 4.4, DFC, Line 192: "a discrminator module and"; **Correction:** "a _discriminator_ module and"
- In Section 4.4.1, Clustering Assignment, Line 199: "can be use as in the Deep Embedding clustering"; **Correction:** "can be _used_ as in the Deep Embedding clustering"
- Formatting of references can be enhanced: IEEE conference on computer vision and pattern recognition -> IEEE Conference on Computer Vision and Pattern Recognition (CVPR).







**Familiar With The Original Paper:**

I have read the original paper

**Reproducibility Summary:**

Report has summary

---

### Decision · Program_Chairs · 2021-03-31

**Decision:**

Reject

**Comment:**

Overall reviews and/or the paper content not good enough for the AC to recommend to the journal.